# Current Imaging Diagnosis of Hepatocellular Carcinoma

**DOI:** 10.3390/cancers14163997

**Published:** 2022-08-18

**Authors:** Evangelos Chartampilas, Vasileios Rafailidis, Vivian Georgopoulou, Georgios Kalarakis, Adam Hatzidakis, Panos Prassopoulos

**Affiliations:** 1Radiology Department, AHEPA University Hospital, Medical School, Aristotle University of Thessaloniki, 54636 Thessaloniki, Greece; 2Radiology Department, Ippokratio General Hospital of Thessaloniki, 54642 Thessaloniki, Greece; 3Department of Diagnostic Radiology, Karolinska University Hospital, 14152 Stockholm, Sweden; 4Department of Clinical Science, Division of Radiology, Intervention and Technology (CLINTEC), Karolinska Institutet, 14152 Stockholm, Sweden; 5Department of Radiology, Medical School, University of Crete, 71500 Heraklion, Greece

**Keywords:** hepatocellular carcinoma (HCC), ultrasound (US), contrast-enhanced ultrasound (CEUS), computed tomography (CT), magnetic resonance imaging (MRI), perfusion imaging, MR diffusion imaging, multiparametric imaging, diagnostic algorithms, locoregional treatment

## Abstract

**Simple Summary:**

The role of imaging in the management of hepatocellular carcinoma (HCC) has significantly evolved and expanded beyond the plain radiological confirmation of the tumor based on the typical appearance in a multiphase contrast-enhanced CT or MRI examination. The introduction of hepatobiliary contrast agents has enabled the diagnosis of hepatocarcinogenesis at earlier stages, while the application of ultrasound contrast agents has drastically upgraded the role of ultrasound in the diagnostic algorithms. Newer quantitative techniques assessing blood perfusion on CT and MRI not only allow earlier diagnosis and confident differentiation from other lesions, but they also provide biomarkers for the evaluation of treatment response. As distinct HCC subtypes are identified, their correlation with specific imaging features holds great promise for estimating tumor aggressiveness and prognosis. This review presents the current role of imaging and underlines its critical role in the successful management of patients with HCC.

**Abstract:**

Hepatocellular carcinoma (HCC) is the fourth leading cause of cancer related death worldwide. Radiology has traditionally played a central role in HCC management, ranging from screening of high-risk patients to non-invasive diagnosis, as well as the evaluation of treatment response and post-treatment follow-up. From liver ultrasonography with or without contrast to dynamic multiple phased CT and dynamic MRI with diffusion protocols, great progress has been achieved in the last decade. Throughout the last few years, pathological, biological, genetic, and immune-chemical analyses have revealed several tumoral subtypes with diverse biological behavior, highlighting the need for the re-evaluation of established radiological methods. Considering these changes, novel methods that provide functional and quantitative parameters in addition to morphological information are increasingly incorporated into modern diagnostic protocols for HCC. In this way, differential diagnosis became even more challenging throughout the last few years. Use of liver specific contrast agents, as well as CT/MRI perfusion techniques, seem to not only allow earlier detection and more accurate characterization of HCC lesions, but also make it possible to predict response to treatment and survival. Nevertheless, several limitations and technical considerations still exist. This review will describe and discuss all these imaging modalities and their advances in the imaging of HCC lesions in cirrhotic and non-cirrhotic livers. Sensitivity and specificity rates, method limitations, and technical considerations will be discussed.

## 1. Introduction

Hepatocellular carcinoma (HCC) is the commonest primary liver tumor, comprising 75–85% of cases. HCC ranks sixth in global incidence after breast, lung, colorectal, prostate, and gastric cancer. In terms of mortality, it ranks third for both genders, and rates of incidence and mortality are 2–3 times higher in men than in women [1]. Although the increasing prevalence of metabolic syndrome has shifted the etiology of liver cancer, hepatitis B and C account for 56% and 20% of global mortality, respectively. Variations by world region exist; for example, alcohol consumption accounts for 22% of all HCC cases in Europe and North America in 2020 [2]. The vast majority of HCCs are diagnosed in patients with cirrhosis and/or chronic hepatitis B infection, necessitating close surveillance in these groups in order to detect the tumor at an early stage.

Imaging plays a key role in surveillance, diagnosis, and staging, as well as post-treatment follow-up. Ultrasound surveillance improves survival in a cost-effective way and is endorsed by all major practice guidelines, i.e., American [3], European [4], and Asian-Pacific [5]. Dynamic contrast-enhanced Computed Tomography (CT) takes advantage of the hemodynamic changes that occur in the cirrhotic nodule as it progresses to early HCC during the multistep process of hepatocarcinogenesis. The gradual decrease in both normal arterial and portal supply and the formation of unpaired arteries are exemplarily exploited in multiphasic examinations after contrast agent administration; the appearance of the suspicious nodule on the acquired images serves as the basis for its characterization and subsequent management decisions. Magnetic resonance imaging (MRI) has the advantage of assessing additional features such as nodule cellularity and presence of fat, which are also important in nodule assessment. The use of hepatospecific contrast media has significantly augmented the diagnostic performance of MRI: as these drugs are taken up by specific transporters, whose expression decreases as carcinogenesis progresses, lesion hypointensity on the hepatobiliary phase not only is a sensitive feature—even allowing detection of high-grade dysplastic nodules—but is useful for predicting histologic differentiation too. Contrast-enhanced ultrasound may characterize tumor hemodynamics with comparative capacity; furthermore, use of a specific sonographic contrast medium can also aid tumor detection via imaging on the late Kupffer phase.

The characteristic tumor hypervascularity on the arterial phase and hypoperfusion on the portal phase constitute the hallmark of HCC diagnosis on dynamic CT or MRI. Clinically-important prognostic features can be derived from the imaging appearance of HCC. Disease staging can also be performed during the examination in order to identify satellite or multifocal lesions, portal vein invasion, or extrahepatic metastases. Additionally, as HCCs are supplied almost exclusively by the hepatic artery, response assessment after locoregional treatment or systemic therapy can be performed based on tumor enhancement on the arterial phase during follow-up CT or MRI scans [6].

In the following paragraphs of this review, a thorough discussion of the imaging characteristics of HCC on the various imaging modalities—including more advanced techniques—as well as a comparison of their diagnostic performance will be provided. A brief comment on the most widely-used diagnostic algorithms is also included—for completeness of the presentation.

## 2. Ultrasound

Liver cirrhosis (LC) is the primary risk factor for HCC, with patients requiring periodical imaging surveillance. Ultrasound (US) is very-well suited for this purpose, thanks to its wide availability, cost-effectiveness, and accuracy in detecting focal liver lesions (FLL). Once a FLL is detected, US can assist its characterization, particularly using the full spectrum of ultrasonographic techniques, including B-mode, colour, and power Doppler techniques, such as pulsed-wave Doppler, non-Doppler flow-visualization, and, currently, contrast-enhanced ultrasound (CEUS); a wide range of techniques justifies the term, multi-parametric ultrasound [7,8].

The appearances of HCC on US vary depending on the size and degree of differentiation. An important distinction in terms of size is the cut-off of 2 cm [9]. On B-mode, HCC is hypo-echoic in more than 50% of cases, although it can be hyperechoic or of mixed echogenicity in approximately 25% of cases, respectively [9]. Consequently, it is important not to misdiagnose a hyperechoic nodule in a cirrhotic liver as a hemangioma and discard it without further investigation. The hyperechoic element of an HCC may represent a fatty component. HCCs smaller than 1 cm can be iso-echoic, and hence difficult to detect. As a general rule, tumor echogenicity reflects cell density. Given the gradual carcinogenesis of HCC inside a cirrhotic liver, an HCC is typically nodular in shape, except for the massive type, which appears irregular. The lesion margins are usually relatively well-circumscribed in the nodular type of HCC, but poorly defined in the massive type [9]. A peripheral hypo-echoic halo may be noted, corresponding to a thin fibrous capsule in 90% of cases [10]. The so called “mosaic pattern” and the “nodule in nodule” appearance are two characteristic demonstrations of HCC in every modality, including US and CEUS [9], which is seen with a frequency proportionally increasing with tumor size. All these B-mode characteristics have been classified into five macroscopic types, as follows: small nodular type with indistinct margins, simple nodular type, simple nodular type with extranodular growth, confluent multinodular type, and infiltrative type, with the potential for malignancy increasing accordingly [11,12,13].

The vascularity of a FLL is important, pointing towards a benign or malignant diagnosis. From a technical point-of-view, colour Doppler is the first-line modality to assess for intratumoral or peripheral vascularity, but it suffers from technical limitations such as Doppler-angle-dependence, low sensitivity to slow flow, and overwriting artifact. Power Doppler and modern non-Doppler flow visualization techniques are now available for improved characterization of HCC vascular architecture. HCCs less than 2 cm commonly appear avascular due to the technique’s low sensitivity, while in some cases blood vessels are visualized as lines or dots inside, or surrounding, the tumor. A continuous waveform on spectral analysis can be seen in these tumors, in keeping with feeding portal flow. Once the tumor increases in size, more characteristic vascular patterns can be appreciated. Namely, the “basket” pattern of vascularity refers to the presence of a fine network of arterial branches surrounding the lesion. Using spectral analysis, both pulsatile and continuous waveforms can be recorded, which correspond to the hepatic artery and hepatic or portal vein origin of blood supply, respectively. In the massive-type HCC, an overall irregular pattern of vascularity can be appreciated [9]. As a general rule, a continuous (portal vein-like) waveform indicates a dysplastic nodule or a well-differentiated HCC; contrarily, a pulsatile arterial waveform is suggestive of advanced HCC [14]. Power Doppler tends to detect intratumoral colour signals in 19% or more of angiographically hypervascular lesions compared to colour Doppler. Tumors appearing hypovascular on angiography typically exhibit no flow signals on either colour or power Doppler. Power Doppler is less affected than colour Doppler by the small size and the deep tumor location [15]. HCCs with a higher resistive and pulsatility index were associated with early recurrence, suggesting a more malignant nature [16].

Doppler limitations have been addressed with newer non-Doppler techniques, like the superb microvascular imaging (SMI, Canon Medical Systems, Otawara, Japan) or b-Flow/high-definition color (HDC, GE Healthcare, Chalfont St. Giles, UK), which exclusively visualizes intratumoral blood vessels without artifacts with high sensitivity, high spatial resolution, and in real-time [9]. When compared to colour Doppler, SMI visualized more signals, while a hypervascular pattern on SMI was significantly more common in HCC, compared with other lesions [17]. A study using SMI on FLL concluded that HCC demonstrates a “diffuse honeycomb” or a non-specific type of vascularity, which is significantly different from hemangioma [18].

Given that patients at risk for HCC formation undergo US surveillance, an important advance was the introduction of US LI-RADS^®^ (Liver Imaging Reporting and Data System), a classification system in accordance with CT/MRI LI-RADS^®^ that was issued by the American College of Radiology. Briefly, this system assesses the quality of examination and the potential of a FLL to represent HCC in three classes and suggests further management. The features taken into account include size and echogenicity [19]. In the setting of HCC screening and surveillance, US LI-RADS^®^ yielded a sensitivity of 58–89% and a specificity >90% [20]. US in general has a reported sensitivity of 98% and specificity of 85% for overall HCC detection [21]. Tumor size is nonetheless a significant factor as the technique’s sensitivity can only be 65% for lesions <2 cm. The same applies to non-alcoholic steatohepatitis, where the overall change of liver echogenicity lowers the sensitivity [22].

## 3. Contrast-Enhanced Ultrasound

A decisive turning point in the ultrasonographic diagnosis of HCC was the introduction of contrast agents in combination with modern techniques like the pulse-inversion technique. These features not only increased the sensitivity and specificity of CEUS in diagnosing HCC, but also augmented the modality’s role in diagnostic algorithms. There are two contrast agents widely used for HCC: SonoVue (Bracco, Milan, Italy) and Sonazoid (GE Healthcare, Amersham, UK). Both agents work with a low mechanical index (MI) and in real-time, but the latter is thought to be phagocytosed by reticuloendothelial (Kuppfer) cells of the liver parenchyma, thus generating a late arterial phase (post-vascular), starting from 10 min, when the liver parenchyma is normally enhanced, whereas malignant lesions appear washed-out due to lack of Kuppfer cells.

In Europe, CEUS is usually performed with SonoVue^®^, which is not uptaken by Kupffer cells and hence produces an arterial, portal-venous, and late arterial phase lasting up to 6 min [23]. The hallmark of HCC on CEUS using SonoVue^®^ is a homogeneous and intense arterial phase hyper-enhancement (APHE) with mild wash-out starting >60 s after injection (Figure 1). Nodules measuring >5 cm may show heterogeneous enhancement due to necrosis (Figure 2). Both the size and degree of differentiation affect the enhancement pattern of HCC [24,25]. Wash-out is less often seen in HCC nodules <2 cm, but is more frequent in HCC with poorer grades of differentiation [23]. The timing and degree of wash-out are important for the characterization of HCC, which typically shows milder hypo-enhancement compared to metastasis and cholangiocarcinoma. HCC washout should start at least 60-s post-injection, while a quarter of cases may become hypo-enhancing 3-min post-injection, justifying the need for prolonged scanning in patients with cirrhosis. Early wash-out (<60 s) has been associated with poorly-differentiated HCC and non-hepatocellular malignancies [23]. An important study published in 2020 looked at CEUS findings of the entire spectrum of carcinogenesis in the cirrhotic liver by examining regenerative nodules, dysplastic nodules, and small HCC. It was concluded that shorter contrast–arrival times in lesions compared with background liver was associated with a higher degree of malignancy [26]. A different study determined 0.5 s as the minimum difference in contrast arrival time between the nodule and liver as a criterion for HCC [27]. On the contrary, arterial phase iso- or hypo-enhancement and late arterial phase iso-enhancement were associated with a lower degree of malignancy [28]. Regenerative and dysplastic nodules generally tend to appear iso-enhancing to adjacent liver cells on all phases of the scan [29] (Figure 3). In a study analyzing HCC ≤3 cm, moderately- and poorly-differentiated HCCs exhibited arterial phase hyperenhancement (APHE) more often than well-differentiated HCCs [30]. Other researchers found that moderately-differentiated HCCs are more commonly hypervascular during the arterial phase, while atypical appearances (iso- or hypo-vascular) may occur more often in well- and poorly-differentiated HCCs. Wash-out time has been found to be shorter in poorly-differentiated HCCs, as compared to well-differentiated lesions [31]. Despite ongoing research, overlap in blood supply origin between different steps of hepatocellular carcinogenesis may still limit the technique’s accuracy in diagnosing HCC [29,32,33]. CEUS advantages and disadvantages are summarized in Table 1. In cases followed-up with CEUS or US, an increase in size or change in echogenicity indicates malignant transformation [27]. According to the European Federation of Societies for Ultrasound in Medicine and Biology (EFSUMB) guidelines, CEUS can be used in cases with inconclusive CT or MRI or if the patient is not suitable for biopsy, as well as for monitoring changes in enhancement in nodules requiring follow-up [23].

Since both post-ablation necrotic areas and malignancies appear non-enhancing on the post-vascular phase, a way to discriminate those is a second dose of Sonazoid while targeting the non-enhancing area; arterial enhancement within the non-enhancing area thus establishes malignancy. This approach appears to outperform contrast-enhanced CT with a sensitivity of 95.4% and an accuracy of 94.7% [34]. CEUS with Sonazoid^®^ has been found to be equivalent or even better than contrast-enhanced CT for the detection of arterial hypervascularity of HCC nodules [35,36,37]. The post-vascular phase can also be assessed using a high-MI Doppler pulse where the disrupted microbubbles within the normal parenchyma generate a signal, whereas the lack of microbubbles inside malignancies is accentuated, appearing in the form of “punched out” areas [9].

Similar to CT and MRI, a CEUS LI-RADS^®^ algorithm has been introduced by the American College of Radiology to aid in the accurate characterization of nodules in liver cirrhosis patients. The major criteria are APHE, nodule size, and late-mild wash-out. A rim APHE or early (<60 s) or marked wash-out represent LR-M criteria, favoring the diagnosis of a non-hepatocellular malignancy. Ancillary features suggesting malignancy include definite growth, while the mosaic architecture and nodule-in-nodule architecture indicate HCC. On the other hand, size reduction or stability for ≥2 years indicates benignity. Classes LR1–2 can return to follow-up, LR-3 needs a second modality sooner than 6 months, while LR-4 and LR-5 require biopsy or treatment [33]. An important difference between CT/MRI and CEUS is that the latter technique does not visualize arterio-portal shunts, meaning that any arterially hyperenhancing lesion represents a true lesion and not a false finding [20]. Upon meta-analysis, CEUS is 93% sensitive and 90% specific in differentiating benign from malignant FLL [38]. In detecting HCC in patients with cirrhosis, a multi-center study showed that category LR-5 has a 98.5% positive predictive value, a 15.5 positive likelihood ratio, but only a 62% sensitivity for diagnosing HCC, making CEUS LR-5 a highly-specific tool [39]. In a study looking exclusively at HCC nodules up to 2 cm, CEUS LR-5 was 73.3% sensitive and 97.1% specific [26].

Temporal Maximum Intensity Projection (MIP) is a useful CEUS technique that generates vascular maps of FLL. A study assessing HCCs with this technique concluded that well-differentiated HCCs exhibited either normal or not-clearly visible intratumoral vasculature. Contrarily, poorly-differentiated HCCs showed tortuous, meandering, or tapering and interrupted intratumoral blood vessels. These parameters were 85% sensitive, 92.7% specific, and 90% accurate [40]. In another study, dysmorphic arteries have been found in 72% of HCCs [31]. The quantification of CEUS signal has also been used to consolidate a qualitative assessment by the performing radiologist, showing significant differences between benign and malignant lesions in terms of parameters such as rise time and late-phase ratio between lesion and liver [41]. This type of analysis, also termed “dynamic CEUS”, is able to detect microvascular invasion of HCC, since various parameters such as wash-in rate and wash-out rate are significantly higher in invasive tumors [42]. Dynamic CEUS also detected higher wash-out in cholangiocarcinoma than HCC by using quantitative parameters, while the arterial enhancement profiles of these tumors were identical [43]. This analysis also has implications in tumor response to anti-angiogenetic treatment where perfusional parameter changes can be monitored [24]. Parametric maps based on the time of microbubble arrival is an alternative and simplistic quantitative approach (Figure 4).

HCC in the non-cirrhotic liver typically appears on CEUS with APHE and wash-out on the portal-venous or late-phase, while a chaotic vascular architecture may be seen. The fibrolamellar variant of HCC has non-specific appearances on CEUS but generally demonstrates rapid heterogeneous wash-in and quick and intense wash-out [23,44].

## 4. CT

Multidetector Computed Tomography (MDCT) plays a pivotal role in the diagnostic work-up of cirrhotic patients who are at increased risk of developing HCC. According to almost all guidelines, recognition of a nodule ≥10 mm by ultrasonography (US) during HCC surveillance should be followed by dynamic MDCT or MRI examination [45].

Nowadays, MDCT is widely-available, rapid, and robust. Most modern CT scanners have the capability to image with wide-detector arrays, typically more than 8-row detectors, allowing large *z*-axis coverage in a single rotation with high spatial resolution [46]. In comparison to MRI, MDCT is a well-tolerated examination, less prone to motion artifacts, even in elderly or non-cooperative patients who are unable to hold their breath. The main disadvantages include radiation exposure and relatively low contrast resolution and tissue differentiation. Moreover, studies have demonstrated a slightly higher sensitivity and specificity of MRI compared to CT, especially for lesions smaller than 20 mm.

Consistent and appropriate CT imaging protocols are absolutely requisite for optimal detection and characterization of liver lesions in a cirrhotic patient, thereby allowing for the reproducibility of LI-RADS categories [47,48]. The Technique Working Group of LI-RADS has proposed minimal technical requirements for the performance of CT in order to achieve wide acceptability and optimal imaging (Table 2).

Multiphase contrast-enhanced imaging consisting of the late arterial, portal-venous, and delayed phase is invaluable for a confident imaging diagnosis of HCC.

The typical imaging hallmark diagnostic feature of HCC is the combination of non-rim arterial hyperenhancement (non-rim APHE) on the late arterial phase and non-peripheral wash-out appearance on the portal-venous and/or delayed phases on MDCT or MRI, thereby reflecting the peculiar vascular derangements that occur during hepatocarcinogenesis [49] (Figure 5). According to the latest EASL/EORTC (European Association for the Study of the Liver/European Organization for Research and Treatment of Cancer) guidelines issued in 2018, a definite diagnosis of HCC can be established in a nodule measuring ≥10 mm or based on a background of liver cirrhosis or other risk factors for HCC, if these typical imaging hallmark features are encountered [4]. Additionally, AASLD (American Association for the Study of Liver Diseases)/LI-RADS guidelines have endorsed two other major imaging features, namely, an enhancing “capsule” depicted on the venous-portal and/or delayed phase, and the threshold growth defined as ≥50% increase in size of a mass over ≤6 months [50,51] (Table 3).

**The non-contrast phase** is mainly suggested at initial diagnosis and in patients who have prior locoregional therapy, as it is essential to distinguish lipiodol staining and blood products from true arterial hyperenhancement.

The **late hepatic arterial phase** is considered the most determinant vascular phase for the assessment of HCC, as APHE is an essential feature of HCC and most current guidelines do not allow a definitive diagnosis of HCC for nodules without it [4,5,46,52].

The late arterial phase is characterized by full hepatic arterial enhancement with good portal vein enhancement, but no antegrade enhancement of the hepatic veins. The early arterial phase, which is characterized by the intense enhancement of the hepatic arteries without enhancement of the portal vein is considered inappropriate for the depiction of hypervascular lesions, as most HCCs are not conspicuous until the late arterial phase. As the late arterial phase occurs during a restricted time interval, using a fixed time delay, imaging can be inaccurate in patients with cardiovascular disease or other co-morbidities. This is the reason, most institutions recommend the use of patient individualized scan protocols such as test-bolus or bolus-tracking methods, with the latter performing better [46] (Table 4).

The **portal-venous phase (PVP)** is characterized by the maximal enhancement of the portal veins, adequate enhancement of the hepatic arteries and the hepatic veins by antegrade flow, and peak enhancement of the hepatic parenchyma. Scan timing for the portal-venous phase is not as restricted as for the late arterial phase and generally occurs 60–80 s after the start of the injection [46]. This phase is essential for visually assessing the reduction in enhancement of HCC nodules relative to the surrounding liver parenchyma, which is a subjective perception designated as “wash-out” or “wash-out appearance”. Several investigators have demonstrated that early and profound “wash-out” is related to higher tumor grade and the probability of microvascular invasion [53,54,55].

The **delayed phase**, also known as the equilibrium phase—due to an equilibrium state between the vascular space and the interstitial space—is obtained >120 s after the start of injection, preferably with a scan delay of 3–5 min [46]. It is characterized by decreased but persistent enhancement of the portal and hepatic veins and liver parenchyma. The use of the delayed phase increases HCC’s conspicuity, particularly in tumors smaller than 2 cm, as in some cases the wash-out appearance is better depicted during that phase. Furthermore, in almost 10% of cases, “wash-out” is observed only in the delayed phase. The delayed phase is also optimal for the detection of enhanced capsular and mosaic appearance, features that are characteristic and relatively specific for progressed HCC.

An important imaging feature included in the AASLD/LI-RADS’ major criteria is the **enhancing “capsule”**, which is visible as a smooth, sharp, and uniformly thick, thereby enhancing the peripheral rim during the portal and, mainly, the delayed phase [49]. The enhancing “capsule” has a specificity of 86–96% for HCC in high-risk patients, and according to the latest LI-RADS v.2018, an observation measuring >2 cm with non-rim APHE and an enhancing “capsule” can be diagnosed definitely as HCC (LR5), even in the absence of “wash-out” [50]. The tumor capsule is found in about 70% of cases and is a characteristic pathologic feature of progressed HCCs, exhibiting expansive growth [49]. It is not observed in dysplastic nodules or early HCCs and does not occur with intrahepatic cholangiocarcinomas (ICC). An intact capsule has been associated with better prognosis as it prohibits tumor cells from embolizing downstream. Several clinical studies have shown that, after adjusting for tumor size and grade, HCCs with intact capsules are related with lower recurrence rates and better overall survival [56].

Hypervascular lesions that may mimic HCC in patients with cirrhosis or chronic liver disease include perfusion alterations, hemangiomas, focal nodular hyperplasia (FNH)-like nodules, high-grade dysplastic nodules (HGDN), small ICCs, combined hepatocellular-cholangiocarcinomas (cHCC-CCAs) and, rarely, hypervascular metastases in a cirrhotic liver [54,57,58,59,60,61] (Table 5).

It is worth noting that at least 40% of HCCs present with atypical imaging features and don’t fulfill the appropriate vascular criteria to be diagnosed as definite HCCs. These generally include small-size HCCs (<2 cm), either early HCCs or well-differentiated HCCs, poorly-differentiated HCCs, progenitor-type HCCs, scirrhous HCCs, and the infiltrative and diffuse types of HCCs [54,62].

Kim et al. [63,64] have shown that approximately 17% of small HCCs can be isodense on the arterial phase and hypodense on the portal-venous phase, whereas about 40–60% of small HCCs, even if hypervascular on the arterial phase, don’t exhibit “wash-out” on the portal phase. This atypical appearance has been linked to the multistep process of hepatocarcinogenesis, during which there may be diminished portal tracks before adequate recruitment of unpaired arteries, or there may be neoangiogenesis without significant loss of portal tracks [65]. Further characterization and differentiation of these nodules requires the application of ancillary imaging features, acquisition of specific MRI sequences, or the application of CEUS, which can provide further information concerning their cellularity, function, and vascularity [65,66,67,68]. Moreover, according to a recent large prospective study, including 296 observations in 240 patients [69], the combination of MDCT and MRI in LI-RADS yielded a better diagnostic performance for HCC than MDCT or MRI alone.

The **ancillary imaging features** that can be readily assessed with MDCT and indicate the presence of HCC mainly include the nodule in nodule architecture, the mosaic appearance, the non-enhancing capsule, and, to a lesser degree, the presence of intralesional fat or blood products [50] (Figure 6). These imaging features strongly favor neoplasia of hepatocellular origin and may be used to distinguish HCC from ICC or metastatic disease, with the exception of cHCC-CCA, which can share the same imaging features with HCC.

The **nodule-in-nodule architecture** reflects the emergence of a progressed HCC within a dysplastic nodule or an early HCC, which results from the clonal expansion of cells displaying less differentiation [70]. The inner nodule corresponding to progressed HCC shows arterial hyperenhancement, while the parent nodule corresponding to well-differentiated dysplastic nodule remains hypo- or iso-attenuated. The nodule-in-nodule appearance is a feature with poor prognostic value, as the inner hyperenhancing nodule has a short volume doubling time (TVDT) and grows rapidly.

Likewise, the **mosaic appearance** is the result of the clonal divergence of cells in various steps of de-differentiation inside a nodule typically larger than 3 cm. The imaging appearance is the reflection of the histology, which is comprised of randomly-distributed nodules or compartments with variable enhancements, separated by irregular enhancing septa and necrotic or hemorrhagic areas or areas with fatty metamorphosis. The mosaic pattern is observed in 28–63% nodules of HCCs [49].

**The non-enhancing capsule** refers to a capsule appearance that is hypodense and not visible as an enhancing rim. It should be unequivocally thicker and more conspicuous than fibrotic tissue around other cirrhotic nodules.

Another important ancillary imaging feature favoring malignancy in general, and not HCC in particular, is the **corona enhancement,** which refers to the enhancement of the venous drainage area of the tumor on the late arterial or early portal-venous phase. It is the result of aberrant and disorganized peritumoral drainage due to the invasion of intranodular hepatic veins, and drainage shifts to the surrounding hepatic sinusoids and, subsequently, to the portal venules [71]. It is seen in up to 80% of progressed HCCs and is strongly associated with microvascular invasion (MVI) and seeding of neoplastic cells, hence most metastatic satellite nodules originate in the peritumoral venous drainage area [71]. MVI is an important prognostic factor of overall survival and of early recurrence after resection, locoregional therapy, or transplantation. MDCT imaging features predicting MVI, apart from peritumoral enhancement, and includes the non-smooth tumor border, nodular rim, capsular disruption, prominent tortuous intratumoral arteries, and large tumor size [72,73].

Recently, Bello et al. [74] brought attention to several HCC subtypes with atypical imaging features and correlated them with their histologic and molecular features; for example, an early peripheral and progressive centripetal enhancement (a pattern similar to ICC) was indicative of the scirrhous subtype. Fowler et al. [75] have also presented distinct morphologic and pathologic subtypes of HCC with different prognostic implications. Pathologic subtypes associated with poor prognosis mainly include macrotrabecular massive HCC (MTM-HCC, presenting with necrosis on imaging), neutrophil-rich HCC (previously described as sarcomatoid subtype of HCC), scirrhous HCC (S-HCC), progenitor-type HCC (expressing stem cell markers such as CK19 or EpCAM), and diffuse- and infiltrative-type of HCC, while subtypes associated with better prognosis, apart from early and small well-differentiated HCC, include clear-cell HCC (CC-HCC), steatohepatitic (showing extensive fatty component), and b-Catenin-mutated HCC (hyperintense on the hepatobiliary phase).

This wide variability of biological behavior among progressed HCCs has been correlated to variable genetic and molecular alterations in the process of hepatocarcinogenesis and has been shown to be related with specific imaging features. Imaging features associated with aggressive biological behavior and poor survival include substantial intratumoral necrosis and targetoid appearance, as well as features suggestive of micro- or macrovascular invasion and bile duct invasion [75,76,77,78] (Figure 7). HCCs presenting with such atypical imaging features are designated according to LI-RADS as LR-M lesions and require biopsy for definitive diagnosis.

A distinct morphologic type is the diffuse or cirrhotomimetic-type, which presents as an ill-defined permeative infiltration of the liver parenchyma with subtle or inconsistent arterial enhancement and heterogeneous “wash-out” [79]. It is frequently associated with invasion of the portal vein (68–100%) and high levels of AFP (>10,000 ng/mL). Due to its reduced conspicuity, diffuse-type HCC is often revealed only when malignant portal vein thrombosis becomes apparent. MDCT features indicating a tumor in the vein (TIV) include heterogenous thrombus enhancement, expansion of the portal vein ≥23 mm, and contiguity to the tumor. One of the most characteristic imaging features is the presence of thrombus neovascularity, which corresponds to the “thread and streak” sign [80] seen during the early arterial phase as a thin linear and chainlike opacities, reflecting the growth of the tumor into the vein (Figure 7).

Finally, as many as 20% of HCCs may involve a non-cirrhotic liver and develop without identifiable histologic precursors (“de novo hepatocarcinogenesis”). Non-cirrhotic HCC has distinct histopathologic and clinical features [81]. It usually presents as a single, large mass (>5 cm) with a mosaic architecture, intratumoral necrosis and fat, and an intact capsule, thereby reflecting the slow, expansile growth of the tumor. Extrahepatic involvement, mainly lymphadenopathy, is more frequently detected and is attributable to the delay in diagnosis. It is predominantly moderate or well differentiated, and despite the large tumor burden at the time of diagnosis, has a better overall survival rate and disease-free survival rate in comparison to HCC in cirrhotic livers.

Despite its limitations, MDCT remains a cornerstone of HCC investigation in patients at risk. Its wide availability and lower cost render MDCT as a first-line imaging modality for the evaluation of suspicious nodules in cirrhotic livers. Moreover, MDCT contributes to the prognostication of patients with HCC by identifying features associated with good or bad prognoses. Inconclusive cases require further investigation with other imaging modalities or histologic verification.

## 5. CT Perfusion

CT liver perfusion (CTLP) is a modern imaging technique that provides quantitative functional information on tissue microcirculation—in addition to morphology—and allows a more comprehensive and reproducible evaluation of focal liver lesions [82,83,84]. During the last decade, CTLP has been extensively studied as an imaging biomarker in hepatocellular carcinoma (HCC) and has a plethora of applications in HCC diagnosis, prognosis, treatment selection, and treatment response assessment [82].

From a technical standpoint, CTLP is based on the analysis of a dynamic CT dataset consisting of sequential images of the liver acquired over time following the injection of IV contrast medium [83,85]. Specialized software is employed to extract functional information from the image dataset by measuring the change of attenuation of liver tissue and reference input vessels over time and generating corresponding attenuation time curves. Perfusion parameters are derived, either by directly fitting the attenuation time curve of each point of liver tissue (model-free approach) or by implementing complex pharmacokinetic models (model-based approach). The results are presented as parametric maps with a color scale (Figure 8 and Figure 9). A variety of pharmacokinetic models have been employed in the past for CTLP analysis, and can differ in the number of inputs, compartments, and fitting method. Nevertheless, most modern commercial applications implement a dual input, dual compartment model using the deconvolution method, which best approximates the perfusion characteristics of the liver. While perfusion parameter names and perfusion analysis models are vendor-specific, most manufacturers provide parameters pertaining to blood flow (BF), blood volume (BV), mean transit time (MTT), and vessel permeability (PS), as well as hepatic arterial blood flow (HaBF), portal liver perfusion (PLP), and their ratio (hepatic arterial fraction—HAF or hepatic perfusion index—HPI). Other perfusion parameters are usually reported in conjunction with MRI perfusion studies (Table 6).

As previously stated, hepatocarcinogenesis is characterized by the formation of unpaired arteries that are not associated with portal vein branches, thereby leading to the presence of an arterioportal blood supply imbalance prior to the development of classic hallmarks of hypervascularity [53,65,94]. Conventional MDCT can accurately detect progressed hypervascular HCC lesions but might mischaracterize small HCC lesions that are associated with incomplete neo-angiogenesis [95]. CTLP can overcome this limitation by separating the hepatic arterial from the portal-venous component of blood flow. Studies in animals with chemically-induced liver tumors have demonstrated a significant increase in HaBF, a decrease in PLP, and a subsequent increase in HAF during the transition of pre-carcinoma lesions to early HCC [96,97]. In addition, progressed HCC lesions exhibited significant changes in perfusion parameters related to hypervascularity (increased BF and BV, decreased MTT) and permeability (increased PS) as expected. These observations have been confirmed in multiple studies in patients [86,87,88,89]. Among available CT perfusion parameters that are based on pharmacokinetic models, Fischer et al. found that a cut-off value of ≥85% HPI exhibited a sensitivity of 100%, while a cut-off value of ≥99% HPI yielded a specificity of 100% for the detection of HCCs in cirrhotic patients (Figure 8, [88]). Recently, Hatzidakis et al. highlighted that even descriptive perfusion parameters, such as the mean slope of increase, might offer high accuracy in discriminating HCC lesions from normal parenchyma (Figure 9, [89]). Other studies have demonstrated differences in CT perfusion parameters between HCC and hemangiomas [98,99,100], liver metastases [101,102,103], or arterioportal shunts [104], indicating that CTLP might be a useful tool for differential diagnosis between HCC and other focal liver lesions (Table 7).

In addition to HCC carcinoma diagnosis, CT perfusion can offer insights into tumor aggressiveness and prognosis. Sahani et al. found significant differences in BF, BV, and PS perfusion values between patients with well-differentiated HCC and other tumor grades [86]. Thaiss et al. revealed a good correlation between BF, BV, and HPI perfusion values with VEGFR-2 expression levels of HCC tumors [109]. Bai et al. showed that CLTP values in the periphery of HCC lesions correlate with microvascular density on pathology, which is an important prognostic factor for HCC [110]. Multiple studies have validated CTLP as a non-invasive tool to predict and assess the response of HCC to locoregional and systemic treatment [111,112,113,114,115,116]. However, this subject remains out of the scope of this review.

Although CTLP has many applications in HCC diagnosis and management, its use remains limited outside large reference centers. Patients may need to undergo an additional examination, which might be a burden in daily production and is associated with additional contrast media and radiation exposure. Generated parametric maps are vendor-specific, and as a result, radiologists need to acquire experience and establish intra-institutional reference values before using CTLP in HCC diagnosis. However, great effort has been made to standardize CTLP acquisition protocols [117] and reduce radiation exposure [118,119,120]. With the advent of comparative studies to indicate when the additional use of CT perfusion might be beneficial for the patient and further development of automated image analysis software, CTLP might reach the availability of brain perfusion, which has a central role in the management of stroke [121].

## 6. MRI

MRI is an excellent modality for lesion detection and characterization thanks to its higher contrast resolution and ability to assess more tissue properties than sole vascularization. According to a recent meta-analysis, the pooled sensitivity and specificity for HCC diagnosis were 70% and 94%, respectively, regardless of tumor size [122]. However, sensitivity is greater for lesions >2 cm (approaching 100%) but drops to 58.3–64.6% for lesions smaller than 2 cm [123,124,125], and it is even lower for sub-centimetre lesions. It is generally agreed, however, that MRI outperforms CT for the diagnosis of HCC smaller than 2 cm, while comparable accuracy is reported for lesions ≥2 cm [124,126,127]. It should be kept in mind that the size of an HCC is a significant prognostic factor and an important criterion in all staging systems. The use of hepatospecific contrast media, namely gadoxetate disodium and gadobenate dimeglumine, increases sensitivity by 5–10% [124,128,129,130].

As mentioned before, the radiological hallmark that enables a confident non-histological diagnosis of HCC is the combination of hypervascularity on the arterial phase and hypoperfusion on the portal phase; as with CT, this “wash-in/wash-out” pattern is indispensable on MRI as well. According to the LI-RADS criteria, no lesion without hyperenhancement on the arterial phase can be definitely characterized as HCC; hyperenhancement has to be “non-rim”, i.e., not predominantly peripheral (in order to differentiate from metastases or cholangiocarcinoma) [51]. However, up to 40% of HCCs show no hypervascularity on the arterial phase, and these mainly represent early or poorly-differentiated HCCs [131,132]. Moreover, 40–60% of small HCCs lack wash-out during the portal phase [133,134] (Figure 10). Additional major and ancillary features are employed to help characterize the lesion and assign a LI-RADS category to it.

As already stated, small lesions (smaller than 1, 1.5, or 2 cm, depending on the publication) more often demonstrate atypical imaging features [135,136]. That is to say that lesions below the threshold of 1 cm cannot be characterized as HCC and follow-up is advised according to both EASL and AASLD guidelines. Small arterially-enhancing lesions may represent arterioportal shunts, perfusion disorders, or small intrahepatic cholangiocarcinomas (which may also show portal wash-out) [60]. Interval increase in size by ≥50% in ≤6 months is a major feature according to LI-RADS (Figure 11). However, it is not accepted by EASL and any lesion growth or change in enhancement pattern not typical of HCC should call for biopsy [4].

The presence of a capsule (Figure 12) is a major finding according to LI-RADS, but not EASL. The capsule is a characteristic feature of progressed HCC and is absent in dysplastic nodules or early HCCs. It shows low T1 and T2 signal intensity and enhancements on the portal and delayed phase at 3 min after contrast injection (or transitional phase if hepatospecific contrast agent is used); on the contrary, corona enhancement occurs earlier on the arterial phase. A capsule should be thicker than the fibrous septa of cirrhosis, which also show delayed enhancement. An intact capsule on imaging has been associated with lower recurrence rates after treatment [137], while extracapsular tumor extension predicts poor survival [138]. It should be stressed, however, that an encapsulated progressed HCC has a worse prognosis than an unencapsulated early HCC; the presence of a capsule confers a better prognosis only when the encapsulated tumor is compared to HCCs of a similar size and grade with breached capsules or without a capsule.

On the T2 sequences, most large HCCs show mild–moderate hyperintensity; in contrast, cirrhotic and dysplastic nodules appear iso-intense or hypo-intense relative to the background liver and early HCCs, which are mostly iso-intense. T2 hyperintensity is attributed to increased arterial and decreased portal vascularity; peliotic changes may also contribute [139,140]. Mildly increased T2 signal intensity is an ancillary—but not specific—feature as it is also observed in other malignant lesions of the liver.

Regenerative nodules, dysplastic nodules, and well-differentiated HCCs may all present with T1 hyperintensity before contrast administration; if subtraction techniques are not used, an erroneous impression of enhancement could ensue or, on the contrary, subtle arterial enhancement could be missed. T1 hyperintensity may be due to the presence of fat, copper, glycogen, hemorrhage, or high protein content. Copper and glycogen tend to decrease as hepatocarcinogenesis progresses [141]; the same is also true for iron, and although siderotic nodules appear hypo-intense on all sequences—particularly the T2*—hyperintensity on the T1 sequences may also be seen. Hepatic iron overload, on the other hand, is predisposed to the development of HCC [142], and the appearance of an iron-free area in an iron-overloaded liver should be regarded as suspicious.

Fatty change is encountered in approximately 40% of early HCCs [143]. With increasing tumor size and histologic grade, fat usually regresses and the percentage drops to 6% in moderately-differentiated HCCs [144], only to increase again in highly de-differentiated tumors. This occurs along with the diminished arterial supply, suggesting a connection between reduced blood flow, hypoxia, and steatogenesis [145]. MRI is superior to CT in detecting fatty change with the use of chemical shift sequences, which show the characteristic signal drop on the opposed-phase compared to the in-phase (Figure 12 and Figure 13). Intratumoral fat can also be used to exclude cholangiocarcinoma, which is also associated with cirrhosis. Nevertheless, the added value of fat identification in a HCC is debatable because, when detected, other more suggestive features (like the vascular pattern) are already present [123].

MRI has the unique capability to assess lesion cellularity, which is translated as the reduced diffusivity of water molecules among the tightly-packed cells of a tumor. Restricted diffusion is an ancillary finding that favors malignancy in general, and not specifically HCC. DWI (Diffusion Weighted Imaging) may fail to detect early HCCs, since their cellular density and architecture do not markedly differ from the surrounding cirrhotic nodules [146]. Higher histological grades are associated with higher DWI signal intensities and corresponding lower ADC (Apparent Diffusion Coefficient) values [147,148]. However, the increased amount of fibrotic tissue in the cirrhotic parenchyma also causes restricted diffusion, reducing the conspicuity of HCC as cirrhosis advances [149]. The addition of DWI to the dynamic contrast enhanced phases or the hepatobiliary phase increases the sensitivity of HCC detection [150,151]. Even as a standalone technique, DWI appears to be an acceptable alternative for HCC diagnosis when a contrast agent is contra-indicated [152].

The introduction of hepatospecific contrast media in the 2000s has opened new perspectives in liver imaging. Following their intravenous administration, multiphasic dynamic imaging is performed—similarly to extracellular agents—and subsequently, they are taken up by functioning hepatocytes via specific transporters (organic anion-transporting polypeptide, OATP). Approximately 30% of high-grade dysplastic nodules demonstrate decreased expression of these transporters; the percentage rises to 70% in early HCC, while all poorly-differentiated tumors show decreased or absent expression of OATP transporters [153], leading to low signal intensity on the hepatobiliary phase. More importantly, the decline of OATP expression precedes the typical vascular changes of hepatocarcinogenesis, making the hypo-intensity on the hepatobiliary phase the most sensitive imaging feature for early diagnosis of HCC [125,126,130,154,155]. Typical HCC appears hypo-intense on the hepatobiliary phase and the degree of hypo-intensity has been correlated to histologic grading [156] [157,158]; however, approximately 10% of HCCs appear iso-intense or hyperintense relative to the surrounding liver on the hepatobiliary phase with gadoxetate (paradoxical uptake). The iso/hyperintensity may not be due to tumor differentiation, but rather represents a peculiar molecular/genetic subtype—probably due to genetic or epigenetic alterations—with less aggressive biological features [159,160]. Additionally, it has been shown lately that hyperintensity during the hepatobiliary phase reflects the activation of the Wnt/β-catenin pathway, which, in turn, is associated with resistance to immunotherapy, thereby suggesting that the specific imaging feature (i.e., paradoxical uptake of gadoxetate) could serve as an imaging biomarker [161,162].

Some HCCs demonstrate a transient rim of enhancement on the late arterial phase, which fades away during the subsequent phases. It corresponds to the draining pathway towards the perinodular sinusoids, a potential route for satellite metastases. It is seen in advanced tumors, which are frequently correlated with microvascular invasion [163,164], and represents an ancillary finding, although not specific for HCC.

When a smaller nodule is seen within a larger nodule, it implies de-differentiation of a cell subpopulation and progression towards hepatocarcinogenesis. The “nodule-in-nodule” sign suggests development of HCC within a dysplastic nodule (Figure 13) and the typical HCC features, such as the wash-in/wash-out pattern or diffusion restriction, are seen in the inner nodule. When numerous foci with different imaging characteristics are seen within a nodule, the appearance is known as a “mosaic” pattern and is usually encountered in large tumors (Figure 12), thereby facilitating the differentiation from cholangiocarcinoma [54].

Overall, MRI, with its superb contrast resolution, ability to assess functional parameters, and use of hepatospecific contrast agents, is the imaging modality of choice for the characterization of a suspicious nodule detected during the screening of high-risk patients. In addition to diagnosis, important prognostic features can be extracted with a direct impact on clinical decisions.

## 7. MR Perfusion

Comparable to CT perfusion, it is possible to calculate parameters related to liver microcirculation from MR images. The most common approach is called Dynamic Contrast Enhanced imaging (DCE imaging) and it is based on the quantification of positive enhancement (the ‘T1 effect’) of the contrast agent over time [165,166]. In a similar fashion to CT perfusion, several T1 weighted DCE-MRI source images are acquired following the injection of gadolinium and are subsequently analyzed with specialized software to extract perfusion-related parameters [167,168]. Most studies employ either in-house developed software to analyze DCE-MRI images or commercially-available solutions that do not incorporate liver-specific kinetic models. In addition, the calculation of quantitative perfusion parameters from the source images is even more challenging because the signal intensity does not have a linear relationship to contrast media concentration. As a result, a direct comparison of results between different studies is not straightforward. Nevertheless, relative comparisons of findings within the same study group are feasible.

Overall, a few DCE-MRI studies have been performed on patients with HCC, with findings in line with those of CTLP. Regarding semiquantitative (descriptive) perfusion parameters, HCCs display a higher area under the curve (AUC), higher wash-in, and positive enhancement integral (PEI), as well as lower wash-out and time to peak (TTP), compared to liver parenchyma (Table 6, [90,91,105]). Wash-in, wash-out, and PEI can be used to differentiate hemangiomas from malignant liver lesions (HCCs and metastases) (Table 7, [105]). Using a variety of dual input perfusion models, several authors have reported a higher arterial fraction (comparable to HAF or HPI) and hepatic arterial blood flow, as well as a lower portal-venous flow and distribution volume (v_e_) in HCC lesions compared to surrounding liver parenchyma [90,92,93,107]. Permeability-related parameters like K^trans^ and K^ep^ seem to be more dependent on the characteristics of the employed pharmacokinetic model, and as such, might contribute less to HCC diagnosis [92]. However, Chen et al. demonstrated a significant correlation between these parameters and tumor proliferation status, histological grade, and microvascular density, which suggests that permeability parameters might offer insights into HCC prognosis [169]. In addition, arterial fraction, MTT, and BV seem to differ between HCC lesions, colorectal liver metastases, and hemangiomas and might aid in the differential diagnosis of focal liver lesions (Table 7, [107,108,169]).

Despite all the advantages related to the absence of radiation exposure, and the possibility to combine DCE-MRI with liver specific contrast agents and other MRI techniques, the application of DCE-MRI for liver imaging was, up until recently, hampered by technical limitations. Whole liver coverage was not feasible due to time constraints. As a result, researchers where obliged to select only an oblique slab of the liver containing the input vessels and the target tumor. Additionally, the lack of standardization of acquisition and analysis methods made image interpretation difficult. Recent technological advances, such as golden-angle radial sparse parallel (GRASP) MRI, have permitted the acquisition of free-breathing, whole-liver 3D images with high temporal resolution and without motion artifacts [165]. These images can be used simultaneously for morphological interpretation and for extracting DCE-MRI perfusion parameters without requiring additional contrast injection or scanning time [170]. With a growing number of modern MRI scanners that support compressed sensing sequences in tertiary centers, it is perhaps a matter of time before the development of MR perfusion surpasses that of CTLP and DCE-MRI and becomes incorporated into standard liver imaging protocols.

## 8. PET/CT

Although 18F-fluorodeoxyglucose Positron Emission Tomography (FDG-PET)/CT is an extremely useful tool in the evaluation of many cancers, it is not routinely used for HCC as it is limited by low sensitivity due to the high physiologic uptake of liver tissue and the variable expression of glucose transporters and glycolytic activity in HCC tumors [171]. FDG accumulates in poorly-differentiated HCCs but not in well-differentiated ones. However, tracers based on choline, which is an important component for the synthesis of membrane lipids, recently showed improved detection rates of well-differentiated HCCs [172]. Dual-tracer PET/CT combining choline and glucose as tracers seems to be promising in the staging of HCC patients [172,173,174], although significant overlap between well- and less-differentiated HCCs renders the characterization of tumors challenging based on dual-tracer PET/CT [175]. In a recent study, β-catenin-activated HCCs demonstrated increased uptake of 18F-choline but not 18F glucose, suggesting a potential imaging biomarker that may guide treatment [176]. In another promising study, a correlation between 18F glucose uptake and the expression of genes regulating metabolism was found, proposing a role of FDG-PET in selecting patients for metabolism-targeted therapy [177].

## 9. Artificial Intelligence

The research on Artificial Intelligence (AI) has greatly expanded in the last few years. The application of AI in HCC imaging has demonstrated promising results regarding differentiation from other lesions, prediction of grading and microvascular invasion, identification of specific molecular profile, prediction of response to treatment or post-operative recurrence, and guidance on treatment selection [178,179,180]. However, validation of these results in larger, prospective, multicenter studies is required in the years to come and before AI proves its clinical utility.

## 10. HCC Diagnostic Algorithms

It is clear that imaging has a decisive role in every step of the HCC course, starting from the screening of high-risk patients; to the non-invasive diagnosis, staging, management decisions; and finally, post-treatment follow-up. Improvement in HCC mortality is partly due to the critical contribution of imaging. As HCC can be confidently diagnosed by imaging without confirmatory biopsy, robust imaged-based guidelines are needed. Numerous diagnostic algorithms have been proposed by many scientific societies and organizations all over the world and they are continuously updated, revised, and ever evolving. Up until 2017, 14 imaging-based algorithms had been published [45]; the list has grown since then. All of the proposed imaging systems aim to set standards regarding performance and interpretation of imaging tests in order to maximize diagnostic accuracy, however, their approach differs to a greater or lesser extent.

A major difference is observed between diagnostic criteria proposed by Eastern and Western societies. Eastern societies, like the Japan Society of Hepatology (JSH) and the Asian Pacific Association for the Study of the Liver (APASL), include MRI with hepatospecific contrast media as a first-line diagnostic tool for HCC. In comparison, hypo-intensity on the hepatobiliary phase is considered an ancillary feature that is suggestive of malignancy in general (not HCC in particular) according to the LI-RADS classification system (endorsed by the AASLD) and is not included at all in the EASL guidelines. As stated earlier, hepatospecific contrast agents increase sensitivity for HCC, however some decrease in specificity is expected, even after excluding mimicking lesions with hepatobiliary phase hypo-intensity, such as hemangiomas and cholangiocarcinomas [181,182,183]. This difference in diagnostic strategies is explained by the differences in treatment practices; as liver transplantation is preferred as a curative therapy in Western countries, maintaining high specificity is of utmost importance for the proper allocation of the scant transplant livers. On the contrary, as locoregional treatments are broadly used in Asia, the highest achievable sensitivity is the defining parameter in imaging. For the same reason the presence of a capsule, which is a highly-specific feature of HCC, is considered a major feature in the LI-RADS categorization but is not included in Asian algorithms. Similarly, the Asian guidelines permit the non-invasive diagnosis of a non-arterially enhancing nodule if it appears hypo-intense on the hepatobiliary phase and hypoechoic on the Kuppfer phase of CEUS using Sonazoid, while arterial hyperenhancement is an absolute prerequisite for the confident diagnosis of HCC in both European and American algorithms.

Even with the strictest adherence to imaging protocols, diagnosis of HCC is not always straightforward. When the results of an examination are inconclusive, re-imaging the patient with a different modality, biopsy, or close follow-up could be selected following careful consideration of all clinical and laboratory data in the multidisciplinary team meeting.

## 11. HCC in Non-Cirrhotic Livers

Although most HCCs arise in the setting of cirrhosis through a multistep process that starts from the dysplastic focus, proceeds through dysplastic nodules, and, finally, to overt HCC, some carcinomas arise “de novo”. This term describes the development of HCC bypassing the stage of cirrhosis and is not synonymous with appearance of HCC in a normal liver, as these tumors usually appear through a background of chronic hepatitis B infection or non-alcoholic steatohepatitis (NASH) or, less commonly, metabolic disorders, use of anabolic steroids, or exposure to aflatoxin B1. The relevance between metabolic syndrome, non-alcoholic fatty liver disease (NAFLD, the hepatic manifestation of metabolic syndrome), and NASH (the first stage of the inflammatory phase of NAFLD) on the one hand and hepatocarcinogenesis on the other has drawn attention worth noting recently. The metabolic syndrome deposition of free fatty acids in the liver leads to oxidative stress, inflammation, dysfunction of mitochondria, and activation of stellate cells. Activated stellate cells not only produce collagen leading to fibrosis, but also a number of cytokines, chemokines, and angiogenic factors that play an important role in HCC development by triggering specific pathways [184,185]. Incidence rates of HCC in cirrhosis range widely from 0.7 to 26 per 1000 person-years, depending on the etiology; the risk is higher in viral hepatitis compared to steatohepatitis [186,187]. The percentage of HCCs that arise in the absence of cirrhosis varies across publications and has been reported to be between 1.7–50.1% [188].

HCCs developing de novo tend to be larger than HCCs in cirrhotic patients, which is probably due to the absence of surveillance [189,190]. A large solitary mass with or without satellite nodules is commonly seen; increased tumor size is associated with inhomogeneity and necrosis (Figure 14). Imaging features do not differ and the wash-in/wash-out pattern is usually seen; however, reduced percentages of portal wash-out have been reported in the setting of NASH [191,192]. Despite the increasing contribution of non-alcoholic fatty liver disease to the burden of HCC, only the Asian-Pacific algorithm advises surveillance for NASH. Likewise, application of LI-RADS in the setting of NASH is not recommended. Although patients without underlying cirrhosis present with more advanced stages of HCC, their survival is better, which is likely due to the preserved underlying liver function [193].

## 12. Imaging Assessment of HCC following Percutaneous Locoregional Therapy

Percutaneous minimally-invasive locoregional treatment of HCC includes transarterial chemoembolization (TACE), transarterial radioembolization (TARE), radiofrequency (RFA), or microwave (MWA) ablation (thermal ablation), as well as cryoablation [194,195,196,197,198,199,200]. Among these, thermal ablation and TACE are the most widely adopted [200]. Treatment selection depends on tumor location and size, number of lesions, liver function, and patients’ general condition [4,201]. Post-treatment imaging follow-up is commonly performed with multiphasic contrast-enhanced MDCT or MRI with the goal being to evaluate tumor response to treatment and to detect recurrent disease elsewhere within the liver [4,202,203,204]. Several HCC-specific radiological response classification systems have been developed, including the modified Response Evaluation Criteria in Solid Tumors (mRECIST) and the LI-RADS-Treatment Response (LI-RADS-TR) [205,206]. Compared to conventional tumor response evaluation criteria, LI-RADS-TR and mRECIST focus on the presence of viable contrast-enhancing HCC tissue rather than changes in total tumor size [207]. The interpretation of radiologic findings after locoregional therapy may be challenging, due to the presence of treatment-induced changes in the peritumoral liver parenchyma, which vary among different therapies [207].

Thermal ablation constitutes a first-line treatment with curative intent in patients with small, very-early and early stage HCCs, alongside surgical excision and orthotopic liver transplantation [4,208]. The principle of ablation is the application of thermal energy via an electrode or antenna placed in the tumor, with subsequent heating of the adjacent tumor cells and induction of necrosis [209]. Usually, a contrast-enhanced CT scan is performed immediately after the ablation procedure in order to confirm a “safety” necrotic margin of at least 0.5 cm around the ablated tumor (Figure 15). Coagulative tumor necrosis appears as a non-enhancing area on contrast-enhanced MDCT or MRI, while residual or recurrent tumors will often manifest as a nodular or irregular tissue with arterial phase hyperenhancement near the ablation zone margin [210,211]. A thin, smooth continuous hypervascular rim around the ablation zone might be apparent up to a few months following the procedure, as a result of the inflammation of the adjacent liver tissue due to thermal injury [207,211]. In addition, peripheral geographic areas of hyperenhancement might appear along the needle tract, which correspond to small arteriovenous shunts caused by the needle puncture [212]. A small residual tumor can escape detection in this early stage, though growth on subsequent follow-up imaging, or the presence of “wash-out”, should raise suspicion for viable malignancy [213]. Although dynamic MDCT is the established method for following patients after thermal ablation, due to its wider availability, MRI might offer superior diagnostic accuracy and sensitivity for detection of recurrent HCC, especially with the use of diffusion-weighted images or liver-specific contrast media [122,214]. Few studies with small sample sizes have evaluated the role of CTLP in exploring the hemodynamic changes of HCC nodules after RFA [112,197,215]. Ippolito et al. reported 14 HCC patients treated with RFA. In the hepatic perfusion (HP), arterial perfusion (AP), and hepatic perfusion index (HPI) maps, residual tumors showed significantly higher values (*p* = 0.012, *p* = 0.018, and *p* = 0.012, respectively) compared to the ablated lesion [215]. In another study, Marquez et al. reported 10 HCC patients treated with RFA and found that HPI maps had a higher accuracy in the prediction of residual tumors [112]. CTLP can be very helpful after RFA as well as after MWA, particularly with the use of mean slope of increase (MSI) perfusion maps, in our experience (Figure 15) [89].

TACE is currently the treatment of choice for intermediate stage HCCs, including multinodular asymptomatic tumors without vascular invasion or extrahepatic spread [4,208]. The rationale of TACE treatment rests in the super-selective delivery of embolization particles (lipiodol or drug-eluting beads) together with a chemotherapeutic drug (usually Epirubicin), thereby aiming to induce complete anoxia of the malignant tissue area. This causes selective necrosis of the tumor lesion, sparing as much normal liver parenchyma as possible [216]. Even though TACE might achieve a complete therapeutic response, it is still considered a palliative method, since small numbers of malignant cells can escape necrosis. Early detection of such areas will warrant a chance for repeat treatment, usually with a second TACE session [217].

Multiphase contrast-enhanced CT and MRI are the most commonly used imaging modalities for the follow-up of patients treated with TACE [218]. In a similar manner to thermal ablation, complete treatment response is suggested by the absence of enhancement in the tumor rather than regression in tumor size [219,220]. A thin, smooth rim of arterial phase hyperenhancement adjacent to the tumor margin corresponds to inflammatory tissue and might persist up to 1 year after TACE [207]. In addition, a geographic region of hyperenhancement might be seen in the corresponding liver segment due to occlusion of the feeding arteries and will eventually regress over time [221]. A residual or recurrent tumor usually presents as a nodular enhancing mass with or without wash-out. However, recurrent tumors can also manifest with weak or absent arterial phase hyperenhancement in previously-treated areas [207]. In cases of conventional TACE with Lipiodol or after use of special hyperdense drug-eluting beads, blooming artifacts of MDCT might hamper the recognition of small viable tumor tissue in MDCT; in such cases, MRI offers a clear diagnostic advantage [222]. Additionally, diffusion-weighted imaging can improve the depiction of residual or recurrent HCCs after TACE, and the ADC value may serve as a quantitative biomarker for treatment response [223].

CTLP has been extensively evaluated as a means for the assessment of treatment response in patients with HCC undergoing TACE [224,225,226,227,228]. Yang et al., in a series of 24 HCC nodules successfully treated with TACE, found a statistically significant decrease in arterial perfusion (AP) and hepatic perfusion index (HPI) values in the treated nodules [225]. Chen et al. categorized treatment response by dividing 38 HCC patients into partial response (PR), stable disease (SD), and progressive disease (PD) groups, according to RECIST criteria, and found significantly reduced AP and blood volume (BV) values in the PR group after TACE in comparison to pre-TACE. In the PD group, AP, BF, and hepatic arterial fraction (HAF) showed a significant increase after TACE [224]. Ippolito et al. reported similar results by comparing the HP, AP, HPI, BV, and TTP maps between successfully treated lesions in the site of deposited iodized oil and viable persistence tumors at CTLP performed 4 weeks after TACE; HP, AP, and HPI values were significantly higher in the residual neoplastic tissue compared to the treated lesion (*p* < 0.0001) [226]. Another two papers validated previous studies, both presenting a significant reduction of AP and HPI in successfully treated lesions [227,228]. Hypervascular findings in AP maps reflect tumor-related neo-angiogenesis of viable tissue and should be considered as the most relevant parameter to assess treatment response after TACE in our experience. Similar results can be obtained with the MSI maps with the additional advantage of increased spatial resolution (Figure 16) [89]. Finally, Su et al. demonstrated that CTLP parameters, in particular the AP, Arterial Perfusion Index (API), and PP, are useful predictors of short-term therapeutic response after TACE with a sensitivity and specificity of 87% and 95%, respectively, and an accuracy of 91% [229].

## 13. Conclusions

The role of imaging in the multidisciplinary approach of patients with HCC is constantly evolving and expanding thanks to the application of hepatobiliary and contrast ultrasound agents, as well as newer quantitative techniques that evaluate blood perfusion on CT and MRI. As a result, earlier and more confident diagnosis of HCC has already been achieved; in addition, new imaging biomarkers are very promising in the prognosis of the biological behavior of the tumor. Hopefully, this review has persuasively demonstrated that imaging of HCC is not only an exciting field of active and intense research, but, most importantly, a critical aspect in the effective and personalized management and treatment of patients.

## Figures and Tables

**Figure 1 cancers-14-03997-f001:**
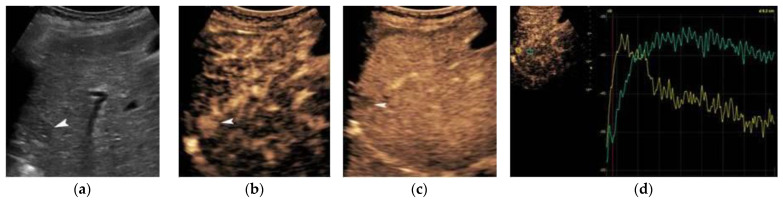
Typical CEUS findings of HCC nodule. B-mode (**a**) showing a hypo-echoic nodule (arrowhead) inside a heterogeneous cirrhotic liver. Arterial (**b**) and portal-venous phase (**c**) CEUS image showing homogeneous arterial phase hyperenhancement and mild wash-out, respectively, in keeping with HCC (arrowhead). Time-Intensity-Curve analysis (**d**) confirming the earlier enhancement of the nodule on arterial phase and wash-out. Quantitative parameters can be extracted using this type of analysis.

**Figure 2 cancers-14-03997-f002:**
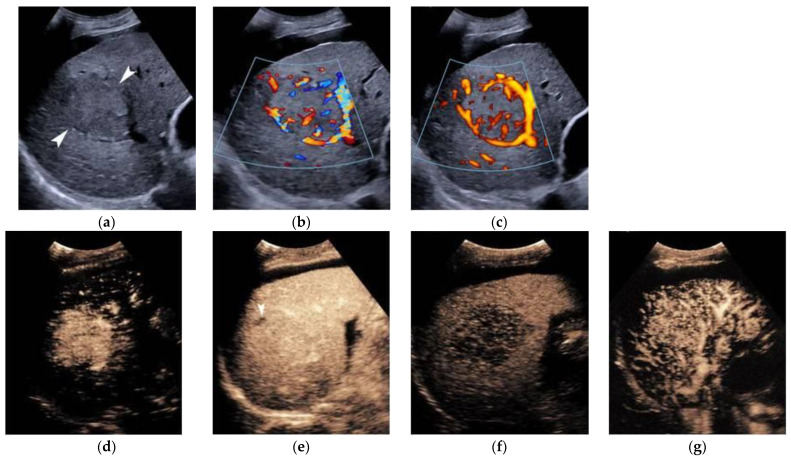
CEUS findings of a large HCC. B-mode (**a**) showing an ill-defined mildly hypo-echoic mass (outlined by arrowheads) inside the right lobe of a cirrhotic liver complicated with ascites. Colour (**b**) and power (**c**) Doppler technique visualizing the irregular internal vascular pattern of the mass. Note the increased vascularity locally. Arterial (**d**), venous (**e**), and delayed (**f**) CEUS image showing arterial phase hyperenhancement prior to adjacent hepatic parenchyma, iso-enhancement on portal-venous phase, and wash-out on the delayed phase (approximately 3 min). Note the area of necrosis appearing as non-enhancing (arrowhead on (**e**)). Temporal MIP image (**g**) showing the vascular architecture of the mass. Note the dense and irregular vascularity indicated inside the lesion.

**Figure 3 cancers-14-03997-f003:**
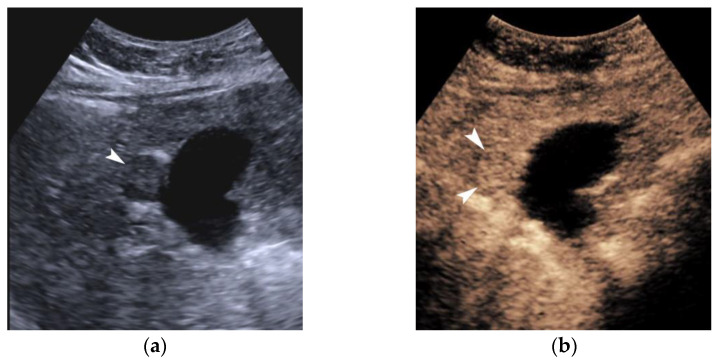
A regenerative nodule in liver cirrhosis. B-mode (**a**) depicted a rounded hypo-echoic nodule (arrowhead) with smooth border. CEUS (**b**) showed that the nodule (outlined by the arrowheads) demonstrated enhancement identical to the adjacent parenchyma with no arterial phase hyperenhancement or wash-out. These findings are in keeping with a regenerative nodule.

**Figure 4 cancers-14-03997-f004:**
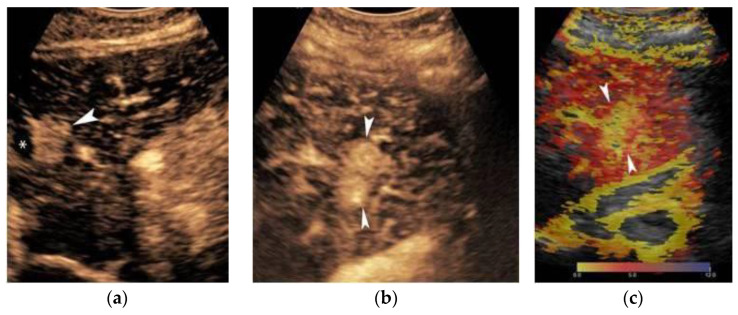
CEUS temporal MIP and parametric image in HCC. Sagittal arterial-phase CEUS image (**a**) showing an HCC nodule (arrowhead) developing next to a previous area of ablation (asterisk). Axial temporal MIP image (**b**) delineating the entire HCC nodule (outlined by arrowheads). Parametric colour map (**c**) confirming the earlier and homogeneous arrival of contrast in the HCC nodule.

**Figure 5 cancers-14-03997-f005:**
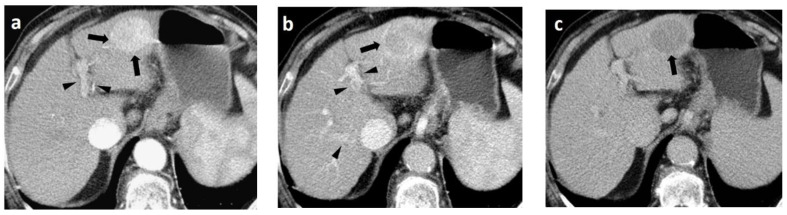
Typical hallmark imaging features of HCC in a 60-year-old patient with HBV cirrhosis. (**a**) Late arterial phase, (**b**) Portal-venous phase, and (**c**) Delayed phase. MDCT images show a 30-mm mass in the left liver lobe with global APHE (arrows) (**a**) “wash-out” on the PVP (arrow) (**b**), and capsule appearance on both the PVP and delayed phase (arrow) (**b**,**c**). Note the enhancement of both the left hepatic artery and portal vein with no enhancement of the hepatic veins on the late arterial phase (arrowheads) (**a**) and enhancement of the left hepatic artery, portal vein, and hepatic veins on PVP (arrowheads) (**b**).

**Figure 6 cancers-14-03997-f006:**
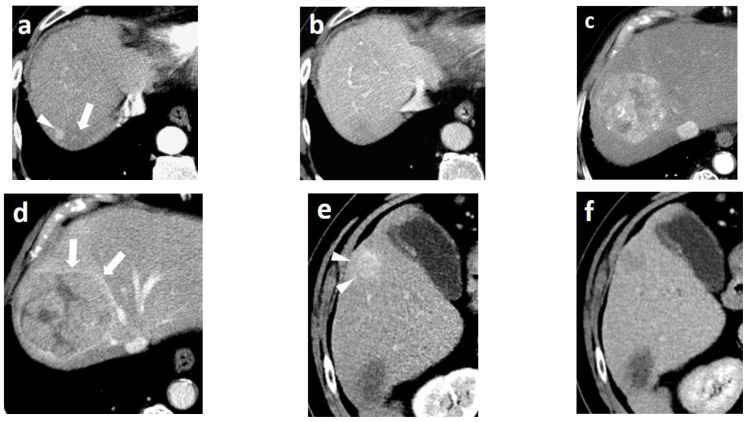
Ancillary imaging features for HCC assessment. Nodule-in-nodule architecture, mosaic architecture, and corona enhancement on the late arterial and portal-venous phase, respectively. MDCT axial images. (**a**,**b**): A small nodule (arrowhead) is located at the periphery inside a larger nodule (arrow), with APHE (**a**) and “wash-out” (**b**) on PVP. The parent nodule is not enhancing. (**c**,**d**): A heterogeneous mass, characterized by enhancing compartments and necrotic areas is seen (**c**). An enhancing capsule is depicted on PVP (**d**) (arrows). (**e**,**f**): There is a transient zone of hyperenhancement around the outer margin of the nodule (**e**) (arrowheads), which fades on PVP (**f**).

**Figure 7 cancers-14-03997-f007:**
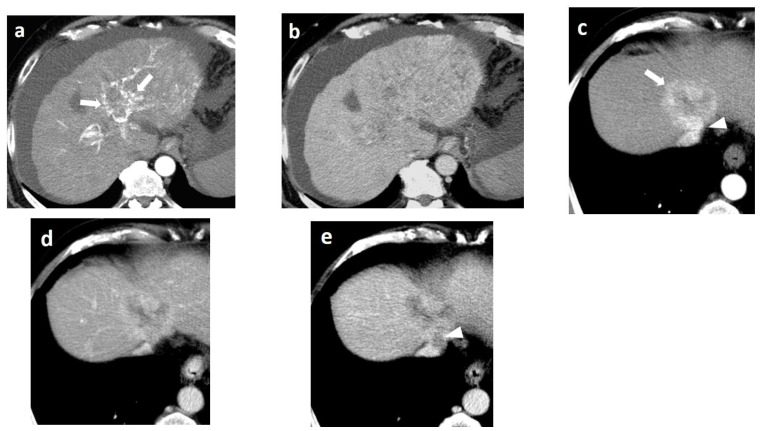
HCC with atypical imaging features; Cirrhotomimetic HCC (**a**,**b**) and Progenitor-type HCC (**c**,**d**,**e**) (late arterial, portal, and delayed phase, respectively). (**a**,**b**): Cirrhotomimetic HCC with TIV. There is marked diffuse heterogenous appearance of the left liver lobe with subtle arterial enhancement and “wash-out” accompanied with hyperenhancing portal vein tumor thrombus, displaying prominent neovascularity (thread and streak sign) (**a**) (arrows). (**c**,**d**,**e**): Progenitor-type HCC. There is a mass with irregular, non-smooth arterial enhancing rim (arrow). Note also the enhancing tissue inside the middle hepatic vein (arrowhead) (**c**) protruding into the IVC (arrowhead) (**e**). With the exception of cHCC-CCA, TIV is consider a fairly specific feature of HCC in a cirrhotic liver since ICC more frequently encases rather than invades veins.

**Figure 8 cancers-14-03997-f008:**
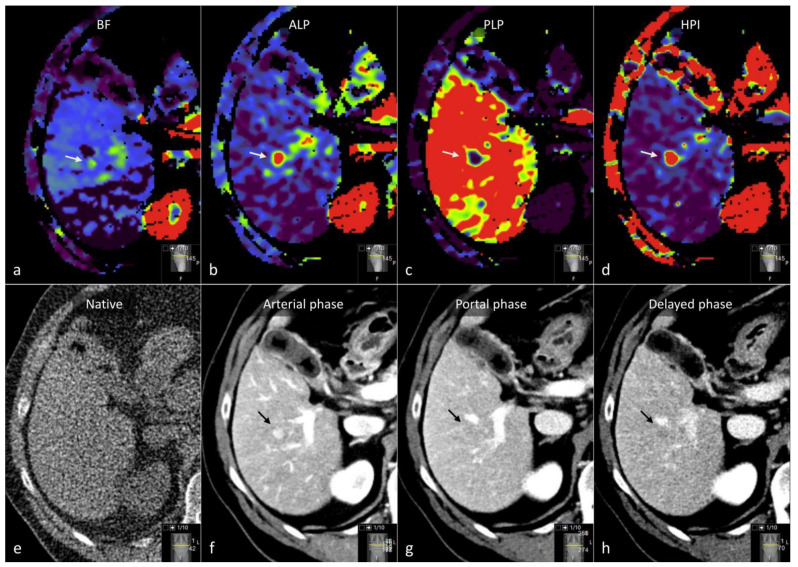
CT perfusion allows the accurate identification and characterization of HCC lesions. In this 69-year-old patient with cirrhosis, a 12-mm nodular lesion in the right liver lobe (arrow) can be easily distinguished from surrounding liver parenchyma on CT perfusion (**a**–**d**), with higher values on the Blood Flow (**a**), Arterial Liver Perfusion (**b**), and Hepatic Perfusion Index (**d**) parametric maps and a lower value on the Portal Venous Perfusion (**c**) map. The lesion is shown on conventional 4-phase CT (**e**–**h**), which was performed on the same day as CT perfusion with arterial phase hyperenhancement (**f**) and wash-out on the portal-venous (**g**) and delayed phase (**h**), which corresponds to LI-RADS 5. BF; Blood Flow, ALP; Arterial Liver Perfusion, PLP; Portal Liver Perfusion, HPI; Hepatic Perfusion Index.

**Figure 9 cancers-14-03997-f009:**
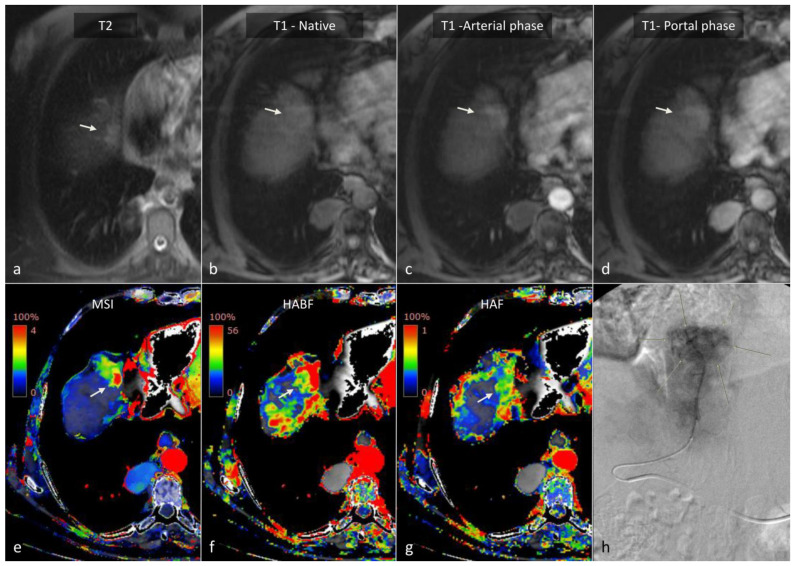
CT Liver Perfusion (CTLP) can complement other imaging modalities for establishing the diagnosis of HCC in difficult cases. This 70-year-old cirrhotic patient was previously treated for HCC with transarterial chemoembolization and presented with a 22-mm subdiaphragmatic lesion in liver segment 8 upon follow-up (arrows). The lesion had a high signal on T2 (**a**) and T1 (**b**) MRI. Contrast enhancement (**c**) and wash-out (**d**) could not be assessed on MRI due to the presence of artifacts. The MSI map of CTLP (**e**) clearly shows avid contrast enhancement in the HCC lesion, which was later confirmed with selective angiography (**h**). Although the Hepatic Arterial Blood Flow (**f**) and Hepatic Arterial Fraction (**g**) parametric maps show high values in the HCC lesion, it cannot be differentiated from surrounding parenchyma due to cirrhosis and prior chemoembolization, which alter normal liver hemodynamics. MSI; Mean Slope of Increase, HABF; Hepatic Arterial Blood Flow, HAF; Hepatic Arterial Fraction.

**Figure 10 cancers-14-03997-f010:**
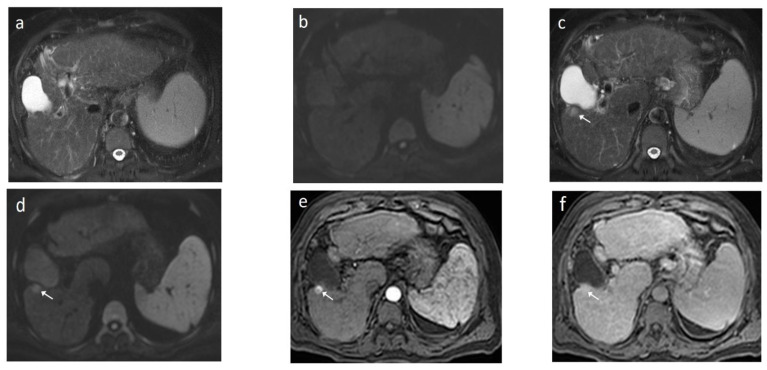
Evolution of a cirrhotic nodule into HCC. No suspicious lesions are identified on the T2 (**a**) and DWI (**b**) sequence of this 66-year-old man with cirrhosis due to hepatitis B infection. On the follow-up scan, performed 3 months later, increased T2 signal (**c**) is now observed in a nodule in segment V, which is associated with diffusion restriction (arrow) (**d**). After contrast administration, arterial enhancements (**e**) without delayed wash-out (**f**) are seen; absence of wash-out is frequent in early HCCs.

**Figure 11 cancers-14-03997-f011:**
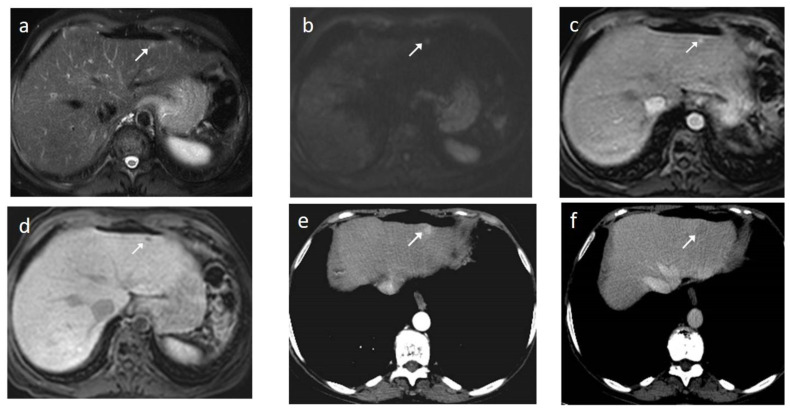
This 48-year-old woman with a history of β-thalassemia major and cirrhosis was followed-up after successful locoregional treatment of two small HCCs. In liver segment II, a 5-mm high T2 signal focus is seen in an anterior subcapsular location (**a**) with associated restricted diffusion (arrow) (**b**). The lesion shows arterial enhancement (**c**) and no uptake of the hepatospecific contrast on the hepatobiliary phase (**d**). Although findings are highly suspicious, the lesion cannot be definitely characterized as HCC, due to its small size. On the subsequent follow up CT, interval growth and typical wash-in/wash-out are now present (**e**,**f**).

**Figure 12 cancers-14-03997-f012:**
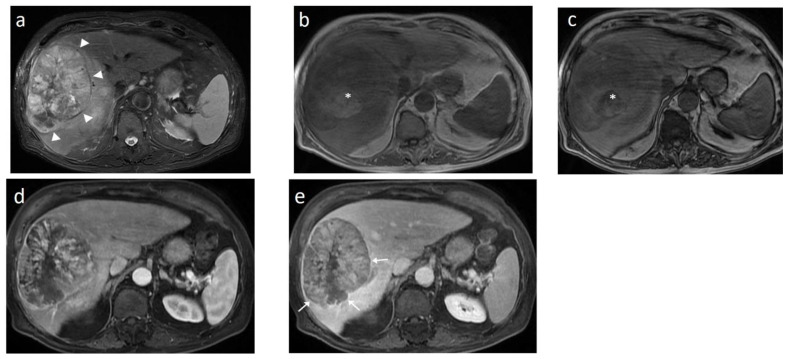
A large HCC is depicted in the right liver lobe of this 81-year-old man. The tumor is surrounded by a capsule, nicely seen as a thin, low signal line on the fat-suppressed T2 sequence (arrowheads) (**a**) and shows inhomogeneous but predominantly high T2 signal intensity. Areas of fat are clearly shown in the in/out of phase images (asterisk) (**b**,**c**). This marked heterogeneity is known as the “mosaic” pattern. After contrast administration, mottled arterial enhancement is noted (**d**); definite wash-out and capsular enhancement (arrows) are seen on the portal phase (**e**).

**Figure 13 cancers-14-03997-f013:**
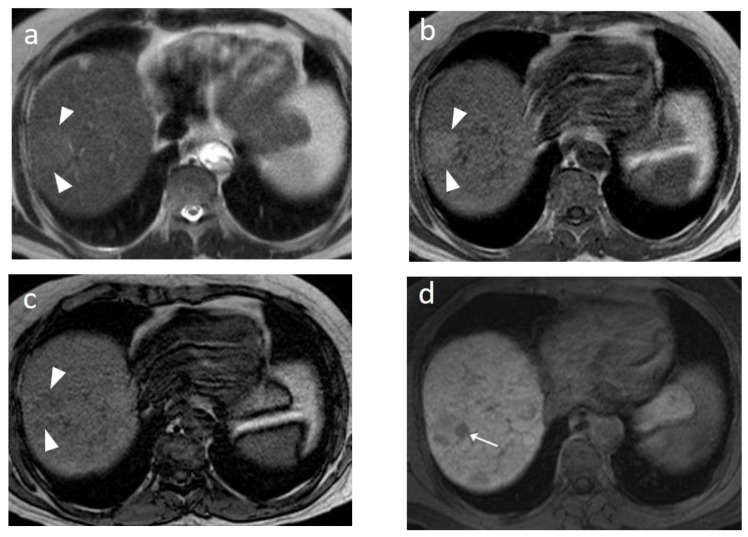
Sixty-five-year-old man with cirrhosis. On the T2 sequence, a nodular high T2 lesion is vaguely seen (arrowheads) (**a**). The lesion appears hyperintense on the in-phase image (**b**) and slightly hypo-intense on the out-of-phase image (arrowheads) (**c**), suggesting the presence of fat. On the hepatobiliary phase after gadoxetic acid administration, a small nodule with markedly decreased signal (no contrast uptake) is evident in the left aspect of the larger lesion (arrow), suggesting focal de-differentiation in a dysplastic fatty nodule and early HCC formation (“nodule in nodule” sign) (**d**).

**Figure 14 cancers-14-03997-f014:**
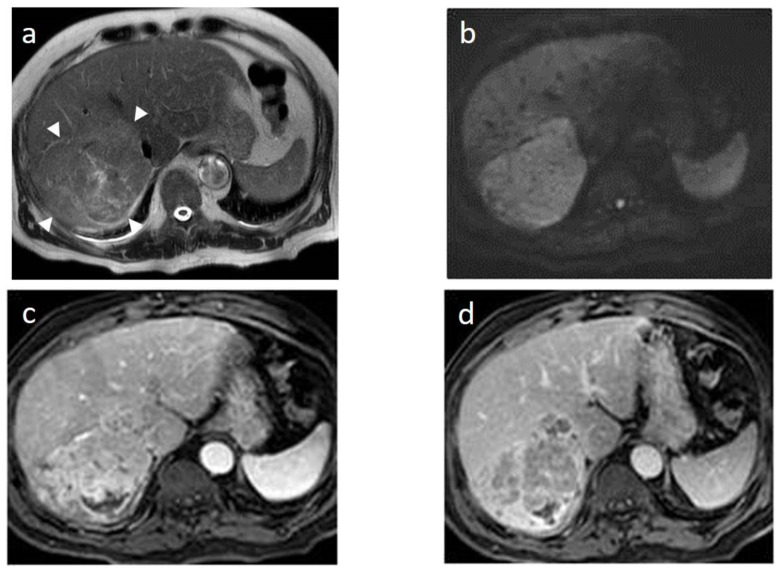
An 11-cm tumor (arrowheads) is incidentally discovered in a 70-year-old man without history of viral hepatitis, non-alcoholic fatty liver disease, or cirrhosis. Otherwise, the lesion shows typical HCC imaging features such as high T2 signal intensity (**a**), diffusion restriction (**b**), arterial hyperenhancement ©, and wash-out during the portal phase (**d**). Due to the absence of risk factors, a biopsy was performed that confirmed the radiological diagnosis.

**Figure 15 cancers-14-03997-f015:**
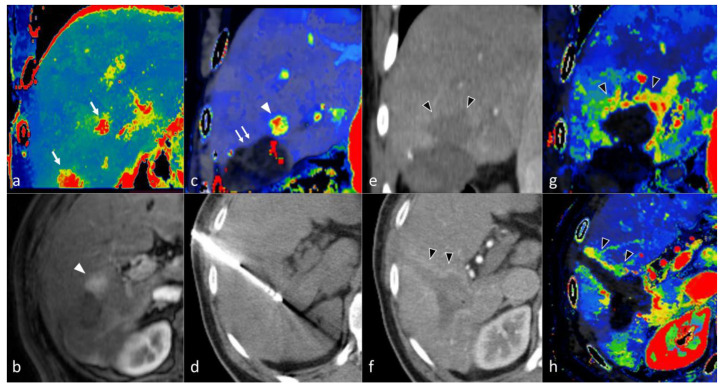
Expected post-treatment changes after microwave ablation (MWA). During US screening examination of a 70-years-old cirrhotic patient, two suspicious nodules, one 2 cm in segment VI and one 1.2 cm more centrally, were found. (**a**) Mean Slope of Increase (MSI) map of CT Liver Perfusion (CTLP) in coronal plane of the same patient shows the two hypervascular lesions in the right liver lobe, both of which proved to be HCCs (arrows). A wedge resection of the inferior tumor was performed. Post-operative arterial phase MRI in axial plane (**b**) and MSI map of CLTP in coronal plane (**c**) revealed once again the centrally-located viable tumor (arrowhead) alongside post-operative changes in the area of the resection (double arrows). A percutaneous MWA of the remaining tumor was subsequently decided. (**d**) Axial non-enhanced periprocedural CT shows the microwave antenna placed inside the tumor. Post-procedural portal phase CT in coronal (**e**) and axial (**f**) plane shows lack of contrast enhancement due to coagulative necrosis in the ablation zone and needle tract with surrounding hyperemia (black arrowheads). Note the safe ablation margin compared to reference images (**b**,**c**). MSI map of follow-up CTLP one month later in coronal (**g**) and axial plane (**h**) shows remaining hyperemia adjacent to the ablation margin, which should not be interpreted as residual or recurrent tumor.

**Figure 16 cancers-14-03997-f016:**
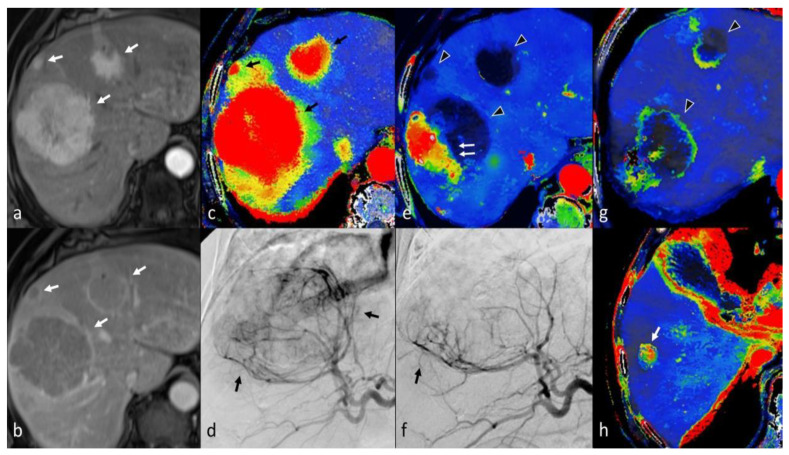
CT Liver perfusion (CTLP) imaging findings before and after transarterial chemoembolization (TACE) of multiple HCC nodules. (**a**,**b**) T1-w MRI scan of 72-year-old cirrhotic patient on arterial (**a**) and portal phase (**b**) after gadolinium injection showing three liver masses with arterial phase hyperenhancement (arrows in a), wash-out, and capsule (arrows in (**b**)) consistent with HCCs. (**c**) Mean slope of increase (MSI) after CTLP confirms MRI findings, showing all hypervascular lesions in red color (black arrows) surrounded by yellow rim, which reflects perfusion disorders. (**d**) Angiographic view of the liver before TACE revealed the malignant hypervascularized overlapping lesions (between black arrows). The smaller subcapsular lesion could not be clearly depicted. The lesions were subsequently treated with selective TACE with Epirubicin-loaded beads. (**e**) Follow-up CTLP one month later (MSI map) revealed extensive necrosis of the chemo-embolized tumors (arrowheads), with a viable area in the dorsolateral portion of the larger mass (double arrows). (**f**) During a second TACE session, the residual malignant tumor lesions were depicted and treated with super-selective TACE. (**g**,**h**). Follow-up CTLP a month after second TACE (MSI map) revealed no signs of viability in the previously found tumors (arrowheads). However, a new, small 1.5-cm lesion had appeared.

**Table 1 cancers-14-03997-t001:** Advantages and disadvantages of CEUS for diagnosis of HCC.

Advantages	Disadvantages
Real-time scanning allowing for optimal detection of APHE with sensitivity Capability to the time of arrival of microbubbles (arterialization) or wash-out; potential biomarkers of malignancy Capability to be performed with free-breathing (good patient tolerability) Non-nephrotoxic agent	Focused field-of-view, hindering staging of the entire liver Operator-dependency Limited by body-habitus

**Table 2 cancers-14-03997-t002:** Basic technical recommendations for CT [46,47,48].

Feature	Recommendation
CT Scanner Configuration	≥8-row multidetector CT
Slice Thickness	2–5 mm
Multiplanar Reformations	Suggested coronal and sagittal planes
Non-contrast Imaging	Suggested for initial diagnosis Required for patients with prior locoregional therapy
Dynamic Contrast-Enhanced Phases	Late Arterial Phase Portal-Venous Delayed Phase
Contrast Administration	≥300 mgI/mL for a dose of 1.5–2 mL/kg body weight (521–647 mgI/kg) Injection rate ≥ 3 mL/s Saline chaser bolus (30–50 mL)

**Table 3 cancers-14-03997-t003:** CT/MRI LI-RADS^®^ v2018 Diagnostic Table by the American College of Radiology (ACR) [51]. LR-3 observations are marked in yellow, LR-4 are marked in orange and LR-5 are marked in red.

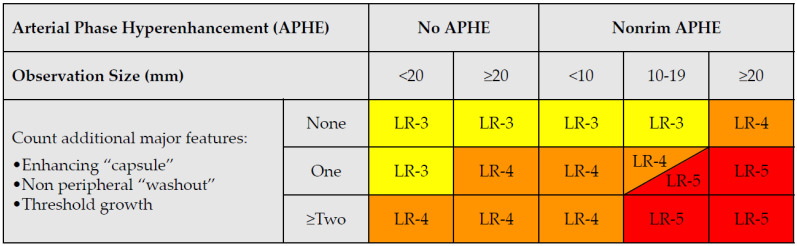

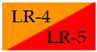
Observations in this cell are categorized based on one additional major feature: LR-4 if enhancing “capsule”LR-5 if nonperipheral “washout” OR threshold growth

**Table 4 cancers-14-03997-t004:** Multiphase Liver CT Imaging and Scan Delay Parameters [46,47,48].

Method	Late Arterial Phase	Portal Venous Phase	Delayed Phase
**Bolus-Tracking** Individualized scan delay	Image acquisition: 10–30 s after aortic threshold density of 100–150 HU	60–80 s after start of injection or 45–60 s after aortic threshold	3–5 min after start of injection
**Test-Bolus** Individualized scan delay	Image acquisition: 10–20 s after peaking aortic enhancement	60–80 s after start of injection	3–5 min after start of injection
**Fixed Scan Delay** Alternatively for young patients with no comorbidities	35–45 s after start of injection	60–80 s after start of injection	3–5 min after start of injection

**Table 5 cancers-14-03997-t005:** Hypervascular HCC mimickers in patients with cirrhosis or chronic liver disease [54,57,58,59,60,61].

Lesion.	Comments	Imaging Features
**Vascular Pseudolesions**	Attributable to arterioportal shunts Particular common in cirrhotic livers	Peripheral, round, or wedge shaped APHE nodules Isodense on PVP
**Hemangiomas**	Rarely encountered Most demonstrate fibrotic involution (“sclerosed” hemangiomas)	“Sclerosed” hemangiomas demonstrate rim APHE Mimic non-HCC malignancies (~6% of LR-M observations)
**FNH-like nodules**	Particularly common in alcoholic cirrhosis SAA-HN-variant is potentially malignant	Nodules with APHE Isodense or “washout” on PVP
**HGDN**	Rarely depicted on MDCT Mimic early HCC	May demonstrate non-rim APHE and become isodense on PVP or depicted only as hypodense nodule on PVP
**ICC**	Comprise 10–15% of cancers in cirrhotic liver	Small ICCs (<3 cm) frequently demonstrate atypical enhancement pattern with global APHE and “washout” or isodensity on PVP
**cHCC-CCA**	Account for <5% of liver cancers	No constant imaging features Commonly have targetoid appearance but may also mimic HCC
**Hypervascular Metastases**	Very rarely encountered due to unfavorable microenvironment & altered portal venous flow	Lesions with APHE Isodense or “washout” on PVP

***Abbreviations***: **APHE**, arterial phase hyperenhancement; **PVP**, portal venous phase; **FNH**, focal nodular hyperplasia; **SAA-HN**, serum amyloid A(SAA)-positive hepatocellular neoplasm; **HCC**, hepatocellular carcinoma; **HGDN**, high grade dysplastic nodule; **ICC**, intrahepatic cholangiocarcinoma; **cHCC-CAA**, combined hepatocellular-cholangiocarcinoma.

**Table 6 cancers-14-03997-t006:** Glossary of commonly-reported CT and MRI perfusion parameters alongside expected changes in HCC lesions relative to surrounding liver parenchyma. Number of arrows indicates the magnitude of difference.

Parameter (Unit)	Definition/Biological Significance	Expected Change in HCC
**CT Liver Perfusion/DCE-MRI** *Perfusion parameters based on pharmacokinetic models (model-based approach)*		[86,87,88,89]/[90,91,92,93]
Blood Flow, or Total Perfusion (mL/100 g/min)	Total flow rate of blood in liver tissue. Reflects hypervascularity.	↑↑
Blood Volume (mL/100 g)	Intravascular blood volume. Reflects hypervascularity.	↑↑
Mean Transit Time (s)	Residence time of contrast agent in tissue. Shorter MTT might suggest hypervascularity and presence of intratumoral arteriovenous shunts.	↓
Hepatic Arterial Blood Flow, or Arterial Liver Perfusion (mL/100 g/min)	Blood flow derived from hepatic artery. High in lesions with a predominant hepatic arterial supply.	↑↑
Portal Liver Perfusion (mL/100 g/min)	Blood flow derived from portal vein. High in normal liver tissue. Low in lesions with a predominant hepatic arterial supply.	↓↓
Hepatic Arterial Fraction, or Hepatic Perfusion Index (%)	Percentage of blood input contributed by hepatic artery. Low in normal liver tissue. Increased in lesions with arterioportal imbalance.	↑↑
Permeability Surface area product (mL/100 g/min)	Reflects leakage rate of blood from vascular into interstitial space. Virtually zero in normal liver parenchyma where fenestrated sinusoids permit free communication between the intravascular and interstitial space. Countable in liver fibrosis and liver tumors.	↑
K^trans^ (s^−1^)	Transfer constant from plasma to interstitial space. Reported in studies that employ single-input dual-compartment models. Related to vessel permeability.	↓/↑
K^ep^ (s^−1^)	Reflux constant from interstitial space to plasma. Reported in studies that employ single-input dual-compartment models. Inverse relation to K^trans^.	↑
v_e_ (%)	Extra-vascular extra-cellular volume fraction. Related to cell density.	↓
*Descriptive perfusion parameters (model-free approach)*		
Area Under the Curve (unitless)	Area under pixel density curve. High in lesions with vivid enhancement.	↑↑
Slope of Increase, or wash-in (s^−1^)	Running average of the slope of the tissue density—time curve *. High in hypervascular tumors with rapid and vivid enhancement.	↑↑
Slope of Decrease, or Wash-out (s^−1^)	The slope of the line connecting the point of maximum enhancement and the last point of the tissue density-time curve *. Low in lesions that display washout.	↓↓
Time to peak (s)	Time interval between onset of afferent vessel enhancement and peak of the tissue density-time curve *. Short in hypervascular lesions with rapid enhancement.	↓↓
Positive Enhancement Integral (%)	The area under the tissue density curve in each tissue voxel, divided by the area under the curve corresponding to a reference vein ROI.	↑

DCE-MRI; Dynamic Contrast-Enhanced MRI, HCC; hepatocellular carcinoma ↑; increased, ↓; decreased, ↓/↑; no difference or conflicting results *; the density-time curve of CT perfusion corresponds to the signal intensity-time curve of DCE-MRI.

**Table 7 cancers-14-03997-t007:** Differential diagnosis between HCC and other common liver lesions using CT Liver Perfusion and dynamic contrast-enhanced MRI parameters. Number of upwards or downwards pointing arrows indicate magnitude of change relative to surrounding liver parenchyma and HCC lesions in comparative studies.

CT Liver Perfusion/ DCE-MRI Parameter	Reported Behavior in Common Focal Liver Lesions				
	HCC	Hemangioma	Hypovascular Metastasis	Hypervascular Metastasis	Arterioportal Shunt
		[98,99,100]/[105,106]	[101,102,103]/[107,108]	[101,102,103]	[104]
Blood Flow, or Total Perfusion	↑↑	↑↑↑	↓	↑↑↑	↑↑
Blood Volume	↑↑	↑↑↑	↑↑	↑↑↑	-
Mean Transit Time	↓	↓↓	↑↑	↓↓↓	-
Hepatic Arterial Blood Flow, or Arterial Liver Perfusion	↑↑	↑↑↑	↑	↑↑ *	↑↑
Portal Liver Perfusion	↓↓	↓	↓	↓↓ *	↓
Hepatic Arterial Fraction, or Hepatic Perfusion Index	↑↑	↑↑↑	↑	↑↑ *	↑↑
Permeability Surface area product	↑	↑↑	↓/↑	↑ *	-
Slope of Increase, or wash-in	↑↑	↑↑↑	-	-	-
Slope of Decrease, or wash-out	↓↓	↓	-	-	-
Positive Enhancement Integral	↑	↑↑↑	-	-	-

DCE-MRI; Dynamic Contrast-Enhanced MRI, HCC; hepatocellular carcinoma, -; not reported, ↑; increased, ↓; decreased, ↓/↑; no difference or conflicting results *; parameters have not been compared to those of HCC lesions.

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
