# Peer review of "Current Imaging Diagnosis of Hepatocellular Carcinoma"

_cancers, 2022, doi:10.3390/cancers14163997_

Round 1
Reviewer 1 Report
This review by Chartampilas et al. draws up a situation report on the current imaging diagnosis of hepatocellular carcinoma. The main issue of this otherwise complete landscape is to ignore the use of Positron-Emission Tomography/CT radionuclide imaging which is currently evolving, with interesting studies showing that it can be used to improve tumor staging and recurrence (Chalaye, J Hepatol 2018; Talbot JN, J Nucl Med 2010) and also to determine the oncogenotype of HCC (Gougelet, Gastroenterol 2019), both being required to improve treatment allocation in patients with hepatocellular carcinoma. These features require at least to add a paragraph concerning these studies and others.
Author Response
We thank you for your comment. We have added a section on PET/CT. Please see on page 34 of the revised manuscript, the new text typed in red.
Reviewer 2 Report
The manuscript is well written and comprehensive.There are only a few minor comments that may enhance the quality of the manuscript
(1)The mortality rate is unclear. In one instance it is written 6th, in another third. Compared to mortality of GBM (universally fatal), pancreatic and a few others its hard to make this statement with certainty. Please elaborate or correct references.
(2)Metabolic disorders (assuming obesity and others) are mentioned but no added description of how they contribute (e.g., NASH)...addition of a table or description will significantly benefit and showcase the data.
(3) Figures showing the relevance of fibrosis and cancer may be beneficial
(4) There are multiple AI-related imaging technologies for processing and modeling. It is great if the authors add their perspective on this and if affects imaging modalities in clinic.
(5) There is a need for conclusive remarks that seems missing
Author Response
Comment 1:
We thank you for your positive position on our manuscript. In order to avoid confusion and according to the reviewer’s suggestion we have rephrased the second sentence of our introduction. Please see the new text on page 2, highlighted in red.
Comments 2 & 3:
Thank you for pointing out the growing interest on the relationship between hepatocarcinogenesis and NASH/metabolic syndrome. According to your proposal, we have added some comments on this topic. Please see pages 36-37 of the revised manuscript (in the section of “HCC in non-cirrhotic livers”), highlighted in red.
Comment 4:
Indeed, research on AI is rapidly and greatly expanding. However, detailed presentation of AI techniques is beyond the scope of this review, which focuses on clinically well established methods and applications. According to your suggestion we have added a short section on AI, on page 35 (typed in red).
Comment 5:
We apologize for the omission. We have added a conclusion (page 42, typed in red).